# Group and Single Housing of Male Mice: Collected Experiences from Research Facilities in Sweden

**DOI:** 10.3390/ani9121010

**Published:** 2019-11-21

**Authors:** Josefina Zidar, Elin M. Weber, Birgit Ewaldsson, Solveig Tjäder, Josefin Lilja, James Mount, Camilla I. Svensson, Emma Svensk, Eva Udén, Elin Törnqvist

**Affiliations:** 1Swedish Centre for Animal Welfare, Swedish University of Agricultural Sciences, 750 07 Uppsala, Sweden; 2Department of Animal Environment and Health, Swedish University of Agricultural Sciences, 532 23 Skara, Sweden; 3Department of Animal Science and Technology, AstraZeneca, 431 83 Mölndal, Sweden; 4Centrum för försöksdjursverksamhet, Uppsala University, 752 37 Uppsala, Sweden; 5Department of Comparative Medicine, Karolinska Institutet, 171 77 Stockholm, Sweden; 6Department of Pharmaceutical Safety, Swedish Medical Products Agency, 751 03 Uppsala, Sweden; 7Department of Physiology and Pharmacology, Center for Molecular Medicine, Karolinska Institutet, 171 77 Stockholm, Sweden; 8The Swedish 3Rs Center, Swedish Board of Agriculture, 551 82 Jönköping, Sweden; 9The Swedish National Committee for the Protection of Animals Used for Scientific Purposes, Swedish Board of Agriculture, 551 82 Jönköping, Sweden; 10Department of Integrative Toxicology, Karolinska Institutet, 171 77 Stockholm, Sweden

**Keywords:** aggression, animal welfare, laboratory animals, male mice, group housing, 3R, single housing

## Abstract

**Simple Summary:**

The mouse is the most commonly used mammal in scientific research, and housed in research facilities around the world. Mice are a social species, but when housing male mice together in a confined environment in the laboratory, aggression is often observed and can be problematic. Fighting or trying to avoid fighting can be stressful. Furthermore, fighting can lead to injuries which can sometimes be fatal. Mouse aggression is therefore a significant welfare problem and has implications on the 3Rs (Replacing, Reducing, and Refining animal use in scientific procedures and education). In this study, we used a survey and workshops to collect the experiences of animal technicians, veterinarians, and researchers at Swedish research animal facilities relating to mouse aggression and what methods of preventing aggression they practice. Both group housing and single-housing as a consequence of aggression was perceived as problematic and stressful for the animals. In line with current recommendations from the literature, participants perceived that aggression occurred less if mice were grouped with litter mates at an early age, that nesting material was transferred at cage cleaning, and disturbance was kept to a minimum. Experience from practice will play a valuable part in developing guidelines for group-housed male mice.

**Abstract:**

Animals used for scientific purposes are protected by EU legislation. Social animals should be kept in stable groups that enable species-typical social behavior and provide individuals with social comfort. However, when group-housing male mice, aggression within the homecage is a common husbandry and welfare problem. Excessive fighting and injuries due to aggression can cause pain and stress, resulting in individuals being euthanized or housed individually. In addition, stress can alter physiological parameters, risking scientific validity and generating larger sample sizes. Mouse aggression, and the consequences thereof, thus opposes the 3R goals of Refining the methods to minimize potential pain and suffering and Reducing the number of animals used. Animal technicians, veterinarians, and scientists using animals have valuable information on how these problems are experienced and handled in practice. We assembled these experiences from laboratory animal facilities in Sweden, mapping problems observed and identifying strategies used to prevent mouse aggression. In line with current literature, less aggression was perceived if mice were grouped before sexual maturity, re-grouping avoided and nesting material transferred at cage cleaning. Preventing aggression will minimize pain and suffering and enable housing of stable groups, leading to more reliable scientific outcomes and is thus of high 3Rs relevance.

## 1. Introduction

The mouse is a social species, and the mammal most frequently used for scientific purposes worldwide. According to EU legislation [1] “Animals, except those which are naturally solitary, shall be socially housed in stable groups of compatible individuals”. Social housing gives animals the opportunity to engage in species-typical social behavior, and social support from conspecifics has been found to markedly improve health outcomes and thus also model quality [2,3,4,5]. However, in practice, group housing of male mice can be challenging. Aggression within the homecage is one of the most common husbandry problems encountered, with wounding being the second most common clinical condition in mice [6]. When groups of mice are observed fighting excessively, or animals are found wounded, groups are generally separated and individuals can be euthanized or housed individually [7]. Not only is aggression a concern for animal welfare; pain, stress, and social isolation can also alter a number of physiological parameters, creating variability and jeopardizing scientific validity [8,9,10,11]. The issue of mouse aggression and housing is thus of high concern from both a legal, ethical, and scientific point of view and is of great importance when considering the 3Rs (to replace, reduce, and refine animal use in scientific procedures and education). Aggression directly impacts several aspects of the 3Rs. Severe fighting contradicts the 3Rs goal of refining methods to minimize potential pain and suffering. Cages of aggressive animals can have more variability [8], which can decrease power and increase sample size, thus contradicting the 3Rs goal of reducing the number of animals used. Also, both breeding programs and experiments often include extra mice to offset those that will be lost to aggression. 

Several studies have identified factors associated with aggression in mice [12,13,14,15] and some general husbandry recommendations have been suggested to decrease levels of aggression. Some examples are to keep mice in small groups (three animals have been suggested), to group animals at an age of 3–4 weeks, to keep siblings or familiar mice together, and to transfer nesting material at cage cleaning [13,16,17]. However, mice are not always kept according to these recommendations, and sometimes mice fight despite being kept according to them, emphasizing that the nature of aggression is complex. 

Traditional experimental studies of aggression have the advantage of isolating and controlling variables for investigation, but they also have some limitations that contribute to the difficulty of implementing results in normal husbandry situations. First of all, they are usually not performed under normal husbandry conditions. In particular, many studies of aggression in mice use resident–intruder tests. These are staged situations that aim to trigger aggression under certain circumstances, and thus not equivalent to homecage aggression [11]. Secondly, mice in experimental studies of aggression are usually only followed during a couple of weeks while aggression in research facilities can develop over longer periods of time. Furthermore, different studies use different strains, vendors, group sizes, cage types, bedding, nesting material, enrichment items, etc., making it even more difficult to interpret and compare results.

The Swedish Board of Agriculture, the authority that is responsible for animal welfare legislation in Sweden, regularly receives requests for exception from the requirement of group-housing male mice, illustrating the reality of the problem. With this background and the fact that it is well known that group housing of male mice can be challenging, the Swedish National Committee for the Protection of Animals used for Scientific Purposes assigned the Swedish 3Rs Center to look closer at the problem. The aim of the project was to get an overview of the extent and consequences of the problem with male mouse aggression and single housing at research facilities in Sweden. The subjective information collected from this project will eventually be combined with data from a systematic literature review (not part of this paper) in order to put together further guidelines for persons planning and conducting research, working in animal facilities, as well as for regulatory bodies. This work was carried out by a working group with members representing laboratory animal veterinarians, ethologists, researchers, chief animal technicians, and 3Rs specialists. 

Within the scope of this publication, we mapped problems with male mice aggression at research facilities in Sweden using a web-based survey, and on-site visits for workshops. The aim was to gather experiences from people working with mice on a daily basis, as well as from people responsible for husbandry protocols, and from researchers running projects. Therefore, we contacted animal technicians, veterinarians, researchers, and facility managers to get an overview of the situation, how they perceive the problem, and potential strategies they have tried or implemented to prevent or solve the problem. In addition, the aim was to highlight the problems with housing of male mice and encourage people working with laboratory mice to engage and act on the problems with both group- and single-housed male mice.

Information gathered from people involved in caring for and using animals gives insight into how the problem is experienced in practice, as well as on how work on preventing and handling problems is undertaken. This experience is rarely collected or published in scientific journals. This valuable information collected from several different facilities, once combined with further information from the literature, will be useful in developing preventative strategies that will work in practice.

Prevention of aggression will result in less stressed animals and more stable groups of healthy animals, leading to reduced variability and more predictable and reliable scientific outcomes. If aggression is reduced, the extra use of animals will also be eliminated. This project is thus of high 3Rs relevance, and by collecting valuable information without using animals, the approach itself complies with the 3Rs principle of minimizing the number of animals used in research.

## 2. Materials and Methods 

In this study, we used a survey and workshops to collect the experiences of animal technicians, veterinarians, and researchers at Swedish animal facilities relating to mouse aggression. A list of all facilities in Sweden with a permit to use mice in research was compiled. From this, we chose to include 10 laboratory animal facilities located at 6 universities (one of which was divided between a university and a university hospital), 2 pharmaceutical or biomedical companies, and 2 government agencies.

Together, these 10 facilities represented 97% of the total use of laboratory mice in Sweden during 2016 (Swedish Board of Agriculture, unpublished). All users of experimental animals must have an animal welfare body (AWB) and one of their main tasks is to promote a good animal welfare within the establishment. We therefore contacted the AWBs and asked them to help us spread information about the workshops and send out the survey within their animal facilities, hence we do not know exactly how many persons received the survey.

### 2.1. Workshops

The working group sent a query to the AWBs and offered to arrange and hold workshops for researchers, animal technicians, and veterinarians to discuss group- and single-housing of male mice at their facilities. To encourage a good atmosphere the workshops were offered either in Swedish or in English and for mixed groups or separated by profession, depending on what the AWBs thought would be the most beneficial approach at the specific facility. Some of the smaller facilities were offered to join the workshops at the larger facilities. Thus, the smaller facilities located at government agencies joined workshops at the universities. All but one facility were able to host or participate in a workshop within the given timeframe. A total of 10 workshops were held at 7 locations in the period, October 2018–March 2019. Participation in a workshop was rewarded with 1.5 continuing professional development (CPD) credits and the participants received a certificate of attendance.

All workshops were conducted in the same manner following the same instructions, but with different persons leading the workshop (all from the working group). During the workshops, we asked the participants to think of all problems that they had experienced when group- or single-housing male mice at their facilities. The participants were then given 10 min to write down their experiences and to separate each problem on a separate post-it note. All notes were then read out aloud by the participants, one note at a time, and taking several turns around the table to present. This encouraged all participants to be active and involved in the task. The workshop leader clustered the post-it notes according to content on a white board. The workshop leader then summarized the different clusters of notes, giving an overview of which areas had been highlighted as problematic by the participants. In the same manner, the participants where then asked to write down their thoughts or experiences and problem-solving strategies used to minimize and prevent problems with group- or single-housing of male mice. They were allowed to write examples of what they had tried at the facility, and whether it worked or not, as well as to write suggestions for problem-solving that they had not tried at the facility but thought could work. After the rounds, the workshop leader again summarized the clusters of notes to give an overview of all aspects that had been discussed. All notes were then computerized and categorized by the project group when compiling notes from all workshops.

For the first brainstorming session on problems, post-it notes were sorted into group- and single-housing, respectively, and subsequently into the following categories: fighting/injuries/dead mice; effect on behavior/physiological parameters; differences between strains; effect on research; effect of the environment; lack of knowledge and awareness; in-house bred/commercially bred/transport; staff and working environment; single mouse left in cage; space limitation; and cost. 

For the brainstorming session on problem solving strategies, post-it notes were sorted into group and single housing respectively, and into following categories: animal husbandry; enrichment/bigger cages; planning and cooperation; breeding/female companion for both group and single housing. Specific categories for group housing were: single housing; effect on research; differences between strains, and for single housing: euthanasia.

### 2.2. Survey

A web-based survey using Google Forms was distributed via email to the AWBs at the 10 Swedish laboratory animal facilities. The AWBs were asked to forward the web-based survey to researchers, animal technicians, and laboratory animal veterinarians at the respective facilities. 

The survey was open from 30 August 2018 to 16 April 2019, available in Swedish and English and divided into three sections (Appendix A). The first section included questions about the respondent, the overall facility as well as the animal housing. The second part involved the practical aspects of keeping animals, and the third part contained questions addressed to researchers only.

In compiling the results of the survey, the answers given in English were translated into Swedish. Thereafter the complete set of answers was analysed as one.

## 3. Results

### 3.1. Workshops

Nine of the 10 approached facilities participated in the workshops. Two smaller facilities joined workshops at larger ones and in three facilities two separate workshops were held, resulting in a total of 10 workshops. Combined, the workshops attracted approximately 150 people working at the different Swedish research facilities. The largest workshop had as many as 50 participants with a mixture of researchers and animal technicians. The smallest workshop consisted of only three researchers at a facility where separate workshops were held for researchers and animal technicians.

In total, 790 post-it notes were collected at the 10 workshops. The notes were evenly distributed between the two brainstorming activities “problem inventory for group and single housing of male mice” and “problem solving for group and single housing of male mice”, with 381 and 409 notes respectively. All comments were characterized as regarding single or group housing, and were subsequently further categorized.

#### 3.1.1. Problem Inventory for Group- and Single-Housed Male Mice

The majority of the 381 comments collected when mapping problems at the workshops, related to group housing (226 comments, Figure 1a), while a third related to single housing (129 comments, Figure 1b). Twenty-six comments were associated with both single and group housing issues, these were not included in the figures and analysis.

For group housing, the largest subgroup with 136 comments was related to observed fighting, injuries, and dead mice (Figure 1a). This subject was only mentioned in six comments for single housing, and five of these were describing fighting as a cause of mice subsequently being housed singly. For group housing, 56 comments, within the category fighting/injuries/dead mice specified when and why fighting was observed (Figure 2a). The most common cause was that specific strains were perceived as being more aggressive and the second most common was fighting in connection to a situation where the animals had been handled in an experiment or when changing cages.

Specific strains, or the notion of differences between strains, were mentioned in 39 comments from the problem inventory workshop sections. The large majority, 37 comments, were characterized as being related to group housing. The most commonly mentioned strain (19 comments) was C57BL/6, which was observed to be a strain where the males were difficult to keep in groups. Other strains, such as BALB/c and CD-1, were mentioned only in a couple of comments, while 15 comments pointed to perceived differences between strains in general. 

The second largest subgroup of comments in relation to problems with group housing was the effect on behavior/physiological parameters, which was also the largest subgroup in categorizing comments in relation to problems with single housing (Figure 1a,b). Within this category, the comments that related specifically to stress or to an effect on physiological parameters were in approximately equal number for group housing and single housing (Figure 2b). Stress, in relation to group housing, was often mentioned in the context of fighting and/or hierarchy, while stress in single housing was more often mentioned in the context of changes in behavior within the cage or when being handled. Aggressiveness and hierarchy was related to group housing, while observed changes in other behaviors such as stereotypies or level of activity was related to single housing.

Among the 129 comments related to problems with single housing, 53 mentioned reasons for single housing males. The most common reasons given were breeding (22 comments) and single housing as a result of the companion’s death or being the only male offspring at weaning (22 comments). Other reasons less frequently mentioned were experimental procedures, fighting, and transport.

At all but one workshop location, comments relating to a negative effect on research were collected (46 comments out of which 36 have been categorized). For both single and group housing, the most frequent comment referred to an effect on research results, including animals in study groups being lost or rendered unusable. Examples of comments within this category were “losing many animals in the studies”, “varying results when measuring glucose”, and “having to exclude mice from the studies due to injuries” in relation to group housing, and “stress from single housing may affect experiments” in relation to single housing. The remaining categorized comments pointed to experiments that require a specific form of housing or that an experimental procedure (such as blood sampling) is affected.

Comments related to the staff or work environment included difficulties in handling mice because of increased aggressiveness or wounds when mice were housed in groups and difficulties in handling mice due to stress when mice were single housed. A fraction of the comments also related to an increased work burden for the animal technicians when many animals were being housed individually. The other notes within this category mostly consisted of comments or questions such as “fighting and subsequent isolation, at what point should they be separated?”; “what can be done to deal with aggressive behavior without separation of animals?”; or “how can we pool animals after isolation?”

#### 3.1.2. Problem Solving for Group- and Single-Housed Male Mice 

The majority of the comments from the problem solving brainstorming sessions were related to group housing, 302 of the 409 comments. Here, only 47 comments related to single housing, while 60 comments related to both or were unspecific (not included in figures and analysis). Out of the 349 comments that relate to group or single housing, 315 have been further categorized (Figure 3a,b).

The most common category, by far, for group housing was animal husbandry, followed by enrichment/bigger cages. Comments regarding animal husbandry, related mostly to minimization of disturbance and stress (Figure 4a). For instance, having few persons handling the mice, not allowing experiments to be performed in the animal room, keeping a calm environment without people passing through and to minimize disturbing sound and smell. Other common comments were to transfer nesting material/enrichment when changing cages and to keep males in smaller groups. Several comments mentioned the use of smaller group sizes without specifying the number of mice, although, when a number was mentioned, the most common was 2–3 per cage.

Within the category enrichment/bigger cages (Figure 3a), most comments in relation to group housing simply suggested to have more enrichment (25 notes). When a specific type of enrichment was mentioned, the comments most frequently concerned houses (16 notes). Using bigger cages was also suggested at 12 instances. Using more enrichment was also mentioned as an intervention for single housing (Figure 3b).

For single housing, the most commonly mentioned ideas of improvements related to careful planning of husbandry practices and cooperation among staff (16 comments, Figure 3b). Among these, the majority were associated with communication or sharing of experiences (Figure 4b) and many touched upon increased and/or improved communication between animal technicians and researchers For group housing the suggestions within the planning and cooperation category (34 comments, Figure 3b) mostly regarded planning of animal husbandry (Figure 4b) with examples such as animal breeding, purchase of animals, euthanasia, and group composition. 

Among the comments from the problem solving and brainstorming sessions, single housing was mentioned as a solution to avoid aggression in group housing while euthanasia was mentioned as a solution to avoid single housing. 

### 3.2. Survey

A total of 95 individual answers were collected from the web-based survey, 86 in Swedish and 9 in English. The respondents can broadly be divided, based on their given role, into 56 animal technicians, 33 researchers, and 6 veterinarians. Almost 60% of the respondents work at a university or university hospital, 35% at a pharmaceutical or biomedical company, and 6% at a government agency.

At universities, the majority of respondents were animal technicians (70%), whereas the majority at the pharmaceutical or biomedical companies were researchers (61%). All respondents stated that their respective facilities housed male mice. Problems with aggression, injuries and stereotypic behavior were observed by almost all respondents (91%). Of those who had observed problems, all had seen fighting, 40% had observed submissive behavior, and 15% had observed stereotypic behavior. Injuries to the tail, the back, and genital organs were also frequently seen (Figure 5a).

Fighting between male mice was observed less than once a week by 41%, while others observed fighting more often or never (Figure 5b). Fighting was most often seen after cage changing (69%) and at sexual maturity (40%; Figure 5c). The majority, (86%) transferred nesting material at the cage change, only 20% indicated that they transferred bedding material.

Forty percent indicated having noticed strains that are more aggressive, and thus more difficult to house in groups, than other strains. Additionally, 46% indicated having noticed strains that were less aggressive and thus easier to keep in groups. The strains most commonly mentioned as aggressive were C57BL/6 (mentioned 13 times), Swiss-derived strains (incl. CD1 and NMRI, mentioned 10 times) and BALB/c (mentioned 8 times). However, BALB/c was also mentioned by 10 respondents, when asked to specify non-aggressive strains. A maximum group size of 5 male mice per cage was stated by 39% of respondents (Figure 6a). The groups were often established using siblings after weaning (mentioned by 86%). Less frequently occurring answers regarding establishment of groups were “after weaning with new group members” (37%) and “grouped at the breeders before shipment” (36%, Figure 6b). Close to half, 48%, indicated that their facility has a maximum age limit for re-grouping of male mice. Nine percent indicated that there is no such limit at their facility and 13% indicated that they simply do not re-group male mice. For those who specified a limit the most common practice (22%) is 21–28 days (Figure 6c).

As one part of the survey, the respondents were presented with a list of actions regarding animal husbandry and asked to rate their effect on group housing of male mice, if undertaken at their facility. One of the more frequently attempted actions, which was also considered to reduce aggression among group housed male mice, was to establish groups prior to sexual maturity. Other actions frequently tried, but with less pronounced effect, were transferring old enrichment to the new cage, transferring old nesting material to the new cage, and to provide extra nesting material (Table 1).

With respect to animal handling, 89% indicated that they lift their mice by the tail (Figure 7a). However, there was also frequent use of the modern methods when lifting mice, 62% indicated that they lift mice in cupped hands, and 44% by use of a tube. When dividing the methods into traditional methods (tail and forceps) and modern methods (cupped hand and tube) more than half (55%) indicated that they use both traditional and modern methods, while about a 10th (11%) indicated use of modern methods only.

The number of handling events prior to experiments, in addition to cage changes, were indicated by 26% as being 2–5 times (Figure 7b). However, never handling the mice in addition to cage changes, or being unaware of whether the mice are being handled or not was equally common, 25 and 26% respectively.

The most frequent reasons for single housing related to individuals who had been aggressive to others, individuals who had been exposed to aggression by others or individuals that had been used for breeding (Figure 8a). About a third (35%) indicated that their respective facility has criteria to decide when to separate male mice in order to avoid aggression or injuries. The criteria given were broadly categorized into visible injuries (53%), fighting (37%), and aggressiveness (13%).

When keeping male mice single-housed, 30% indicated the use of extra environmental enrichment, as compared to when group-housing males, while 40% do not use extra enrichment. For those who use extra enrichment, the most common forms specified were nesting material (87%), houses (52%), tunnels (29%), and chewing sticks (26%).

The 31 researchers who answered the survey and described their research area represent a broad range of subjects from basic research, to drug development and medical research on disease models and mechanisms in, e.g., cancer, diabetes, and neurological disease.

The researchers more often indicated seeing problems with group housing compared to single housing (Figure 8b). Specific problems observed regarding group housing often included effects on results and that data is lost or could not be used. Fighting and injuries were also frequently mentioned. Regarding single housing the most common answer was that mice do not want to be alone, that it is unnatural and stressful. That single housing affects the results and causes practical problems for the researcher was also frequently mentioned.

## 4. Discussion

Aggression among group housed male mice is a common problem in laboratory animal facilities worldwide. Stress, injuries and death following aggression, make this an important animal welfare issue with direct implications on the 3Rs. As a consequence of aggression, groups of male mice are frequently split into smaller groups, single-housed or euthanized [7]. Sometimes, male mice are also single-housed to prevent aggression. However, single housing of social animals can be stressful and to systematically house mice individually is non-compliant with Directive 2010/63/EU [1]. From a practical perspective, keeping individuals alone is also costly and increases workload. Better routines for housing male mice in groups might help to avoid or decrease aggressive interactions. Staff working in laboratory animal facilities deal with these issues on a daily basis and their accumulated knowledge is an important part in finding strategies of avoiding aggression when group housing male mice.

In this study, we assembled experience about group and single-housing of male mice from animal technicians, researchers, and veterinarians working at 10 laboratory animal facilities in Sweden. These facilities represent 97% of all mice that were held as experimental animals in Sweden 2016. To gather information, we conducted workshops at the facilities and sent out an online survey via the animal welfare bodies. The facilities included both companies, universities, and government agencies that keep male mice in the laboratory, enabling us to get more detailed information about different practices at these facilities in Sweden.

We were successful in conducting workshops with participants from all but one facility and overall the participants appreciated a forum for discussion of the topic across different professions working to various degree with the research animals. Many of the participants expressed an aspiration to have similar forums for discussions about this, or other topics, in the future. Interestingly, although we offered to hold workshops separated by profession, many of the facilities chose to have mixed groups, suggesting a good working climate. The main reason for having separate workshops for researchers and animal technicians, was language barriers, where technicians preferred to have the workshop in Swedish, while researchers sometimes where not native Swedish speakers and workshops therefore held in English. Another reason mentioned was to facilitate an environment where everyone could contribute to discussions without having a hierarchal structure that might inhibit free opinion.

The main difference between the workshops and the survey was that the workshops had open questions for the participants to discuss freely, while the survey consisted of questions with a number of alternatives to choose from to facilitate the analysis of responses. The workshops thus encouraged the participants to freely formulate the problems that they had encountered at their animal facilities as well as strategies they had tested to minimize aggression among male mice. Furthermore, when participants were asked to write down ways of reducing problems with group or single housed male mice, they were allowed to write down both what had been tried, what had worked and what had not worked. They were also allowed to present possible suggestions to handle the problems. Unfortunately, when analysing these comments it was not always clear whether the suggested example was something that had been tried or if it was a suggestion of what might work, therefore we cannot separate these comments. Nevertheless, the overall picture that was obtained relating to why and when mice fight, and what can be done to prevent aggression was very similar between workshops and the survey. The survey was however anonymous, so we do not know how many of the workshop participants had also answered the survey. 

The number of veterinarians that responded to the survey was too low to conclude anything substantial about their responses as a group. This is not surprising since there are only a few veterinarians employed at each animal facility. Therefore, comparisons in this study are always between researchers and animal technicians. There was also a slight discrepancy between respondents at the different facilities. Respondents from companies were mainly researchers whereas respondents from universities were mainly animal technicians. Therefore, when comparing responses between companies and universities, we indirectly have a division between researchers and technicians and cannot exclude the possibility that the answers reflect differences between these work locations, rather than between different professions. However, it might also be a matter of definition, many technicians at companies have some research related tasks included in their work description and might therefore have the title researcher.

### 4.1. Group vs. Single Housing

In line with what has previously been reported [10,11,12,16,17], this study confirms in both workshops and the survey, that aggression is a common problem across animal facilities also in Sweden. This is also clear considering that Swedish authorities have received many requests for permission to deviate from present legislation and keep male mice single-housed. During the workshops, the majority of the comments related to problems with group-housing. Across all workshops fighting, injuries and effects on behavior and physiological parameters were the most commonly reported problems when group housing male mice. Similarly, all survey respondents that had experienced problems with aggression had also observed fighting. 

Single housing was mentioned as a solution to prevent aggression in group-housed males. However, during workshops and in the survey, effects on both behavior and physiological parameters was also raised as a problem when housing males individually. This highlights that problems with stress are experienced both when housing male mice in groups and individually. Single housing might seem like a good solution to prevent aggression, and to reduce stress for the individual mice, after all, no one likes to find injured animals in a cage. However, it is important to remember that the negative consequences of aggression are easier to see since they are visible to the eye. Consequences of single housing are more difficult to grasp, since the effects are rarely something we can see. However, social animals will be affected by being singly housed [2,3,4,18,19], and problems with single housing of laboratory mice have also been reported in the literature; for example, when it comes to changes in different physiological parameters [10,20,21]. Euthanasia was mentioned as another solution to avoid single housing of male mice. However, systematically euthanizing mice that end up alone for various reasons will increase the number of animals used in the laboratory animal facility, and is thus not in line with the 3R principles.

### 4.2. When Were Mice Observed Fighting?

During workshops, male mice kept in groups were reported to fight more after sexual maturity and with increasing age. They were also reported to fight after being temporarily removed from their homecage and after transport. Effects on aggression related to different husbandry routines has also been reported in the literature [12,15,22]. This was further confirmed by the responses to the survey where fighting was reported to be common after cage changing and at sexual maturity.

### 4.3. Strain Differences

It is well known that mice differ in a number of characteristics between strains. Both workshop attendees and survey respondents had experienced strain differences in relation to aggression and C57BL/6 was perceived as being one of the more aggressive strains. This confirms results from another study were survey respondents considered C57BL/6 to be one of the strains too aggressive to group house [10]. However, in a recent study where data was systematically collected, C57BL/6 was among the strains showing low levels of aggression [12]. C57BL/6 is one of the most commonly used strains, which might skew the perception since this strain is encountered more often and this could explain the high number of comments related to this strain. The strain BALB/c was equally regarded as both an aggressive and non-aggressive strain among survey respondents, suggesting that there can also be large variation within a single strain and between research facilities. The supplier has previously been shown to influence prevalence of aggression [12], and we could also see this in our study. However, in both this study and the study by Lidster et al. [12], many facilities used several suppliers and the information was not collected for each specific strain, but this would be interesting to investigate further.

### 4.4. How to Avoid Aggression?

#### 4.4.1. Animal Husbandry

Routines related to animal husbandry was by far the most mentioned comment related to strategies for preventing aggression during the workshops. Examples mentioned were minimizing disturbance and stress, and keeping mice in smaller groups. To keep mice in smaller groups is in accordance with one of the reported recommendations for how to prevent aggression [16,17]. 

#### 4.4.2. Nesting Material, Bedding, and Enrichment

Transferring nesting material and enrichments to the new cage when cleaning cages and adding additional enrichments (especially more nesting material and places to hide) were other suggestions discussed during the workshops. Transferring used, but unsoiled nesting material at cage cleaning, is another recommendation on how to decrease aggression [12,16,17], but the benefits when it comes to transferring enrichment items remains ambiguous [10,11,12]. Interestingly, in the survey, facilities that had tried to transfer old enrichments and nesting material most often reported only slightly reduced aggression (slightly over 30%, whereas slightly over 15% reported considerable reduction). 

Also, the majority of the survey respondents transferred nesting material during cage cleaning. Interestingly, some also transferred bedding material, and transferring soiled bedding has been reported to increase aggression [15]. However, there was not a big difference in how often aggressive interactions are observed by respondents who transferred bedding material compared with those who reported to not transfer bedding material. One explanation could be that only the bedding material that is contaminated with urine will increase aggression, whereas clean bedding material might have similar effect as transferring nesting material between cages. However, to elucidate this further, empirical testing would be required. 

#### 4.4.3. Handling

Previous research has shown that the way mice are handled has a large impact on the subsequent behavior of the animals [23,24,25]. The traditional methods of lifting the mouse by the tail or by the tail with forceps is stressful for mice, whereas the modern method with lifting with a cupped hand or in a tube instead reduces anxiety [24,25]. It is also known that stressful situations might lead to aggression in animals [26,27]; therefore, we included a survey question concerning handling method. Despite the known negative effects of traditional handling, only a small number of the survey respondents used only the new methods, while the majority still use traditional methods or a combination of new and traditional methods. Changing from traditional to new handling methods could potentially reduce aggression, but this needs to be further investigated in a controlled study. It would be interesting to explore why the modern methods are not being implemented fully at Swedish facilities. Unfortunately, the reason for using traditional methods cannot be interpreted from the responses. There was no difference in handling methods between researchers and animal technicians. This might suggest a culture at the facility rather than it being linked to a profession. 

#### 4.4.4. Other Suggestions

Other suggestions commonly mentioned to minimize aggression were grouping males at weaning or buying groups that stay together and using larger cages. Several of these actions are also recommended in the literature [11,12,16,17]. The responses to the survey suggest that grouping males at weaning either with siblings or new group members is common practice at Swedish facilities. Furthermore, grouping males before sexual maturity was reported to considerably reduce aggression, suggesting that the age when grouping is performed can be an important aspect to consider in order to avoid aggression. 

To compensate for housing males individually, adding additional enrichment was the single most common suggestion. Unless there are specific, contraindicating reasons related to experimental design, it is surprising that the addition of extra enrichment is not standard practice since current Swedish legislation [28] recommends providing additional enrichment if mice are housed individually.

### 4.5. Communication

A better dialog between researchers and animal technicians was suggested as an important strategy of improving group housing and to avoid single housing. It was mentioned during workshop discussions that animals sometimes are kept singly longer than necessary. For example, when the animals are no longer needed when experiments are cancelled, changed, or finished, animal technicians are not always informed, which clearly illustrates a lack of communication between researchers and animal technicians. In addition, it was suggested by many project participants that improved planning of breeding and euthanasia could help to avoid single housing. Improving the overall culture of care at the facilities could improve the communication between the different professions working with and caring for the animals. It has also been suggested in the literature that changing routines can reduce the number of singly housed male mice [29].

## 5. Conclusions

There is a clear interest among the employees at the Swedish animal facilities in the problems connected to housing male mice at laboratories, and many workshop participants expressed a gratitude to be given a forum to discuss the topic and share their experience. To find ways to involve persons working directly with the animals is thus important, and we encourage this also when it comes to other issues of laboratory animal husbandry.

It is interesting that many of the recommendations on how to prevent aggression found in the literature are similar to solutions identified in this study, that appear to work in practice—such as grouping mice before sexual maturity, keeping groups together, and transferring nesting material at cage cleaning. However, more studies are needed to look further into why these recommendations are not always followed, and under which circumstances mice still fight. 

The data from the workshops and the survey should not be viewed as scientific evidence, rather the collected data gives a picture of the experience and knowledge found among people working in animal facilities. This data can give us a good idea of the current practices and which of these strategies are experienced to be most successful in reducing aggression. It is obvious that housing male laboratory mice in groups is challenging, but problems are also apparent when it comes to single-housing. Replacing one problem for another is neither a good solution for animal welfare nor scientific validity, and this highlights the importance for clearer husbandry guidelines for both group- and single-housed mice.

Together with current information from the literature, this study will be an important complement for developing guidelines on how to prevent aggression when male mice are housed in groups.

## Figures and Tables

**Figure 1 animals-09-01010-f001:**
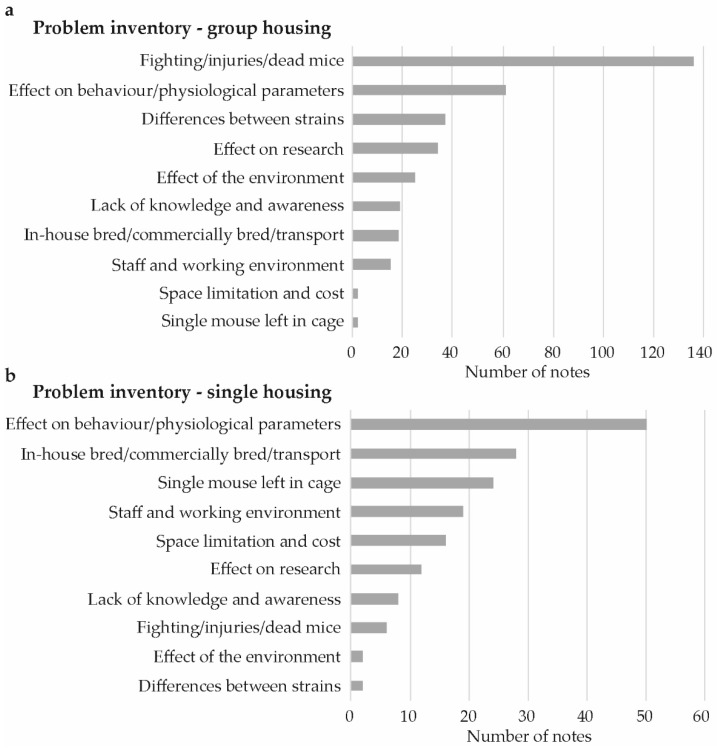
Problem inventory for group- and single-housed male mice. Number of comments per category, describing problems encountered by laboratory animal veterinarians, researchers and animal technicians when: (**a**) Group or, (**b**) Single housing male mice at research facilities in Sweden.

**Figure 2 animals-09-01010-f002:**
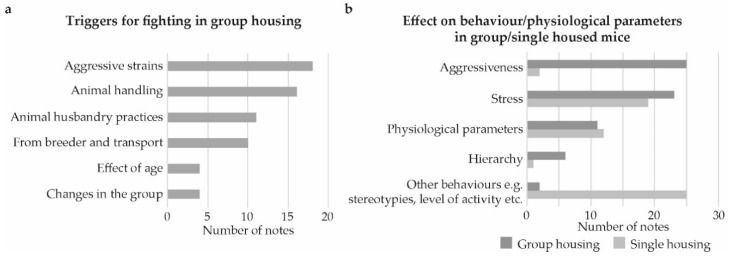
Problem inventory for group and single housing of male mice. Numbers of comments further categorized: (**a**) Describing situations and triggers for fighting when group housing male mice and, (**b**) further categorization of comments related to an effect on behavior/physiological parameters for both group and single housed male mice.

**Figure 3 animals-09-01010-f003:**
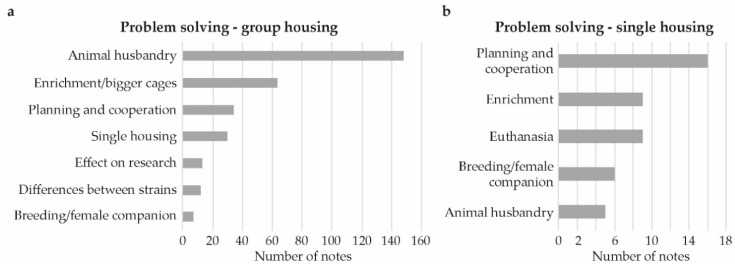
Problem solving for group- and single-housed male mice. Number of comments per category, describing experiences with problem solving or ideas for problem solving by laboratory animal veterinarians, researchers, and animal technicians when: (**a**) group or, (**b**) single housing male mice at research facilities in Sweden.

**Figure 4 animals-09-01010-f004:**
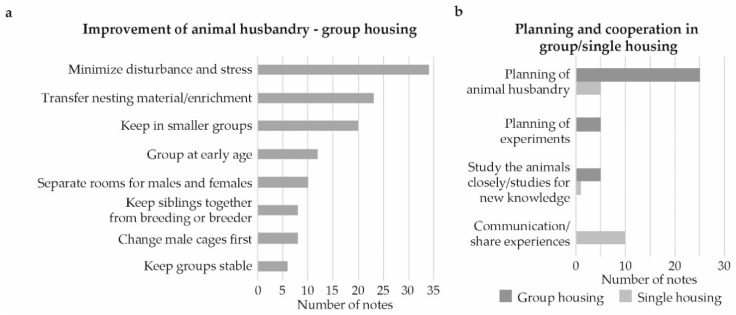
Problem solving for group and single housing of male mice. Number of comments per subcategory for problem solving for: (**a**) Group housing of male mice by certain routines or improvements in animal husbandry and, (**b**) Subcategories for planning and cooperation for both group and single housing.

**Figure 5 animals-09-01010-f005:**
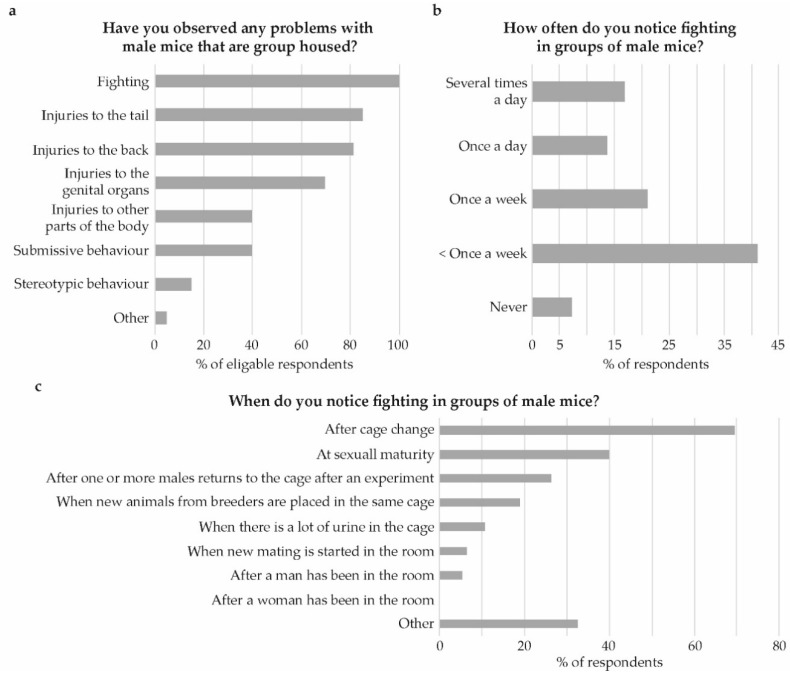
Observation of aggression among group housed male mice. (**a**) Have you observed any problems with male mice that are group housed? (**b**) How often do you notice fighting within male groups? And, (**c**) When do you notice fighting in groups of male mice?

**Figure 6 animals-09-01010-f006:**
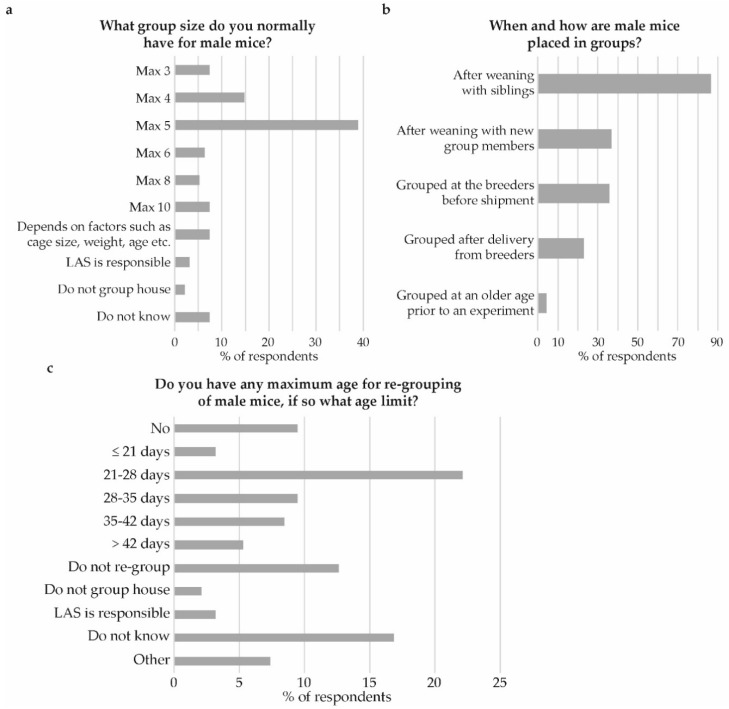
Establishment of male groups. (**a**) What group size do you normally have for male mice? (**b**) When and how are male mice placed in groups? And, (**c**) Do you have any maximum age for re-grouping of male mice, if so what age limit? Note: in panel (**b**) the wording used in the survey was “grouped at the breeders before delivery”.

**Figure 7 animals-09-01010-f007:**
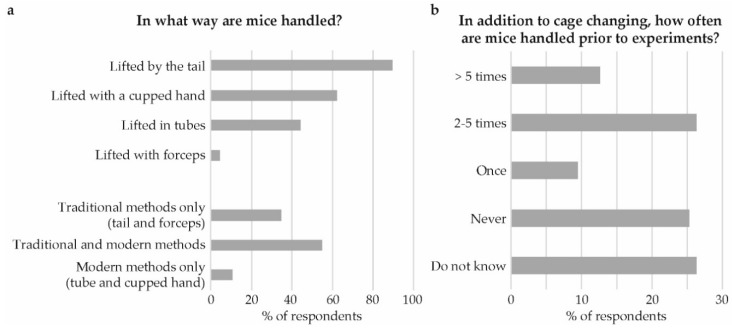
Animal handling. (**a**) In what way are mice handled? And, (**b**) in addition to cage change, how often are mice handled prior to experiments?

**Figure 8 animals-09-01010-f008:**
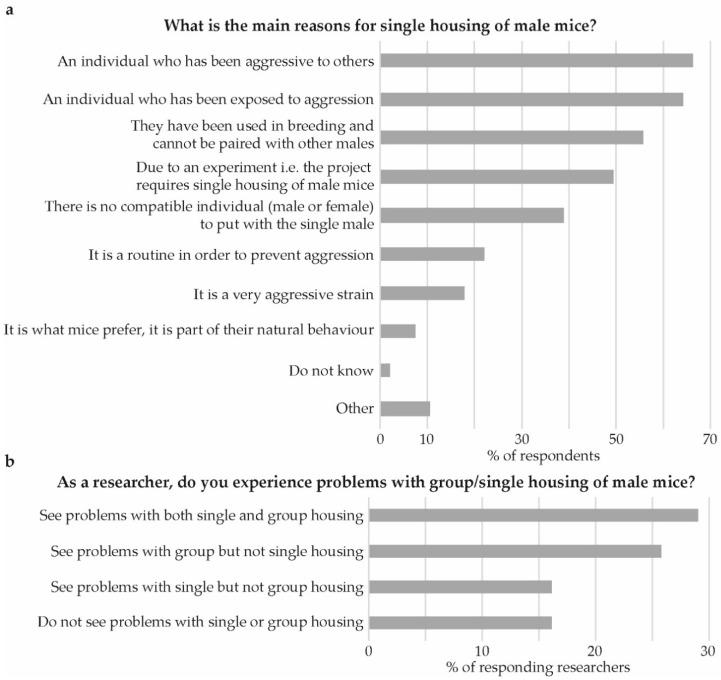
Reasons for single housing and problems seen by researchers. (**a**) What is the main reasons for single housing of male mice? And, (**b**) as a researcher, do you experience problems with group/single housing of male mice?

**Table 1 animals-09-01010-t001:** Attempted actions to facilitate group housing of male mice. Given is the number of answers within each category.

Effect on Aggression							
Attempted Actions	Considerably Reduced	Slightly Reduced	No Effect	Slightly Increased	Considerably Increased	Not Tested	Do Not Know
Establish groups before sexual maturity	35	17	13	0	0	11	19
Transfer old nesting material to new cages	15	31	14	4	0	5	26
Transfer old enrichment to new cages	15	30	17	2	0	6	25
Provide extra nesting material	13	28	20	0	0	8	26
Provide a house/shelter	9	32	12	4	3	6	29
Provide tunnels	1	24	17	1	3	24	25
Provide additional environmental enrichment	3	15	14	0	1	35	27
Handle males and females separately	8	9	11	0	0	39	28
Avoid enrichment that causes competition	5	12	8	0	0	41	29
Use special routines during experimental procedures	9	5	9	0	0	43	29
Transfer old bedding to new cages	2	12	15	2	2	38	24
Provide chewing sticks	1	11	31	1	1	21	29
Enable food-seeking behavior to some degree	0	12	15	1	1	40	26
Provide a shelf/balcony	2	5	5	0	1	59	23
Provide an exercise wheel	2	3	5	0	3	60	22
Provide scented enrichment (a scent to prevent fighting)	2	3	2	0	0	65	23

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
