# Peer review of "Group and Single Housing of Male Mice: Collected Experiences from Research Facilities in Sweden"

_animals, 2019, doi:10.3390/ani9121010_

Round 1

Reviewer 1 Report

1.The paper is very well written and the topic describes a problem which is a common problem all over the world. 

2. Line 298 It is a pity that it was not recorded how many surveys were sent out in order to have an idea about the % of respondents

3. Line 320 Och should be and? 

4. Line 473 Could the large variation in aggression within a strain also be due to differences in the strain from different breeders? 

5. Line 489 It is indeed important to emphasize that when the (nest) material transferred to clean cages is soiled by urine, it probably might increase fighting

6. Line 496 It might have been interesting to know how the way of restraint might effect stress and fighting instead of only handling. 

Reviewer 2 Report

Comment to the Authors:

Addressing individual and group housing in mice in research institutions in Sweden as you do for the first time in your study is an important animal welfare issue. Avoiding fights and bite injuries is an applied refinement and contributes to the culture of care, but I disagree that it complies with the 3R´s principle of minimizing the number of animals used in research. Furthermore, so far there are no scientific studies on the severity of individually kept male mice in comparison to group housing. It is possible that single housing for male mice is more species-appropriate than group housing; this point should be included in your discussion and what is known about the influence of scientific data. Please include the short report from Azkona and Caballero in your introduction and discussion. I therefore recommended publication after minor revision.
